# The Effect of Trans Fatty Acids on Human Health: Regulation and Consumption Patterns

**DOI:** 10.3390/foods10102452

**Published:** 2021-10-14

**Authors:** Davit Pipoyan, Stella Stepanyan, Seda Stepanyan, Meline Beglaryan, Lara Costantini, Romina Molinari, Nicolò Merendino

**Affiliations:** 1Center for Ecological-Noosphere Studies of NAS RA, Abovyan 68, Yerevan 0025, Armenia; david.pipoyan@cens.am (D.P.); stella.stepanyan96@gmail.com (S.S.); seda.stepanyan@cens.am (S.S.); meline.beglaryan@cens.am (M.B.); 2Department of Ecological and Biological Sciences (DEB), Tuscia University, Largo dell’Università snc, 01100 Viterbo, Italy; lara.cost@unitus.it (L.C.); rominamolinari@libero.it (R.M.)

**Keywords:** trans fatty acids, human, ruminal, industrial, gut microbiota

## Abstract

Health effects of trans fatty acids (TFAs) on human organisms can vary according to their type, structure, composition, and origin. Even though the adverse health effects of industrial TFAs (iTFAs) have been widely discussed, the health effects of natural TFAs (nTFAs) are still questionable. Hence, it is important to review the literature and provide an overall picture on the health effects of different TFAs coming from industrial and ruminant sources, underlining those types that have adverse health effects as well as suggesting methods for reducing their harmful effects. Multiple databases (PubMed, Medline, Cochrane Library, etc.) were searched with the key words “trans fatty acid sources”, “ruminant”, “industrial”, “conjugated trans linoleic acid”, “human”, “coronary heart disease”, “cancer”, etc. Reference lists of the studies were scanned discussing the health effects of iTFAs and nTFAs. The review of the literature showed that iTFAs are found to be more harmful than ruminant-produced nTFAs. Although several beneficial effects (such as reduced risk of diabetes) for nTFAs have been observed, they should be used with caution. Since during labeling it is usually not mentioned whether the TFAs contained in food are of industrial or natural origin, the general suggestion is to reduce their consumption.

## 1. Introduction

Dietary fats, including TFAs, have been one of the central topics of discussion in scientific literature and have received more attention from health professionals and the public than any other nutrients in the food supply [1]. TFAs are unsaturated fatty acids containing at least one double bond in its trans configuration [2]. Trans fat is the final product of a chemical process called partial hydrogenation of cis-unsaturated fatty acids.

Even though there is a slight difference in the geometry of the carbon chain, the chemical structure of TFAs is similar to that of saturated fatty acids (SFAs). The question is whether this geometric difference leads to a difference in health effects. The impact of SFAs on human health has been widely discussed in the literature [3,4,5,6], while there is no definite answer on whether TFAs have a negative or positive health effects, as it is highly dependent on their source, type, and transformations they undergo in the organism. It has been shown by randomized controlled trials that, in general, replacing dietary SFAs with unsaturated fatty acids improves cardiovascular health [4]. However, when discussing the health effects of unsaturated fatty acids, an important distinction should be made between its different types, especially TFAs. The health influence of different types of trans fats shall be discussed separately, as they might not only be different, but actually contrasting.

Overall, there are four main sources of TFAs in the human diet; industrially produced TFAs by partial hydrogenation of vegetable oils, TFAs produced during heat processes, TFAs occurring naturally in ruminant sources, as well as TFAs synthesized for utilization as dietary supplements [7]. TFAs are classified according to the two main sources they come from, “industrial” and “natural”. Industrial or artificial TFAs are produced during manufacturing by partial hydrogenation of liquid vegetable or fish oils containing unsaturated fatty acids. On the other hand, nTFAs are produced in the rumen of ruminant animals like cows, sheep, and goats by bacterial transformation of unsaturated fatty acids derived from feed. A small amount of TFAs is also present in poultry and pork fat [8,9,10].

During the reaction of partial hydrogenation, the unsaturated fat is transformed to trans fat, as some of the cis double bonds are converted into trans-double bonds by isomerization reaction. While artificial TFAs are generated through partial hydrogenation with hydrogen gas and a metal catalyst, nTFAs arise through partial hydrogenation with hydrogen produced during oxidation of substrates and bacterial enzymes as catalysts [11,12].

Generally, ruminant animal origin foods contain low quantities of nTFAs and high levels in the diets mainly come from iTFA [13,14,15]. nTFAs are found in animal products like milk, butter, cheese, and beef [12]. Artificial TFAs are eventually found in spreads like margarine, cooking fats for deep frying, and shortening for baking, as well as some popular processed and frozen foods like microwave popcorn, frozen pizza, and snack foods. Huang et al. mentioned that bread and bakery products, cereal and grain products, and confectionery are the top three food groups that contain specific ingredients indicative of iTFA [14]. It is important to note that some TFAs are also formed during the frying of oils at high temperatures. According to Chen et al., the previous studies indicated that several factors including frying conditions, the fatty acid composition, and coexisted antioxidants in frying materials would contribute to the variation in the thermal TFAs accumulation [16].

Both iTFAs and rTFAs consist of the same positional trans isomers, but they differ in distribution and amount [13]. iTFAs contain trans isomers of oleic acid (C18:1 cis-9), the major one being elaidic acid (C18:1 trans-9). In contrast to this, the predominant nTFA in milk and meat is vaccenic acid (VA) (C18:1 trans-11) [17]. In addition, ruminant fat contains small amounts of conjugated linoleic acid (CLA) isomers (C18:2 cis-9, C18:2 trans-11) [18]. Chemical structures of fatty acids are presented in Figure 1 [17].

The health effects of TFAs are very different and highly debated. iTFAs raise LDL (low-density lipoprotein) cholesterol levels and lower HDL (high-density lipoprotein) cholesterol levels, consequently, increasing the risk of coronary heart disease [19]. Moreover, it has been found that consumption of iTFAs increases the risk of atherosclerosis and that TFAs induce apoptosis and inflammation [20]. However, some beneficial effects of TFAs have also been discussed, such as the anticancer properties of animal-origin CLA as well as its ability to fight inflammation and reduce the risk of cardiovascular disease [21,22]. Indeed, in the systematic evaluation of Fuke and Nornberg, CLAs have a potential to be used as functional ingredients for the prevention and control of several chronic metabolic disorders [23]. Nevertheless, in the comprehensive review by Benjamin et al. [24], accumulated clinical evidence (regarding cardiovascular health, immunity, asthma, cancer and diabetes, oxidative stress, insulin resistance, and irritation of the intestinal tract) indicates that there is no consistent result regarding the safety and efficacy of CLA. CLA is not eliciting significantly promising and consistent health effects to uphold it as either a functional or a medical food [24]. Yet, there are studies stating that the health effects of conjugated fatty acids are not yet fully demonstrated in humans. Therefore, clinical evidence appears to be insufficient regarding the effects of CLA supplements on body fat reduction as it depends on isomer type and dose [25,26].

In this review, not only the possible health effects of different types of TFAs but also dietary intakes and worldwide regulatory procedures are presented and discussed.

## 2. Objectives and Methods

Despite a plethora of studies discussing the various effects of TFAs (LDL/HDL cholesterol ratio, progression of cell apoptosis, relation to immune response, cancer, etc.), there is no definite conclusion on whether nTFAs have similar adverse effects as iTFAs. Some of the adverse health effects of TFAs, such as cardiovascular disease (CVD) risk, are very well known and documented; however, there is little research showing whether the health effect of TFAs is related to its type, source, and composition [27]. Thus, there is a need to thoroughly analyze the literature and obtain an overall picture on the health influences of TFAs considering their origin and the possible transformations they undergo in organisms. It is particularly important to understand whether both nTFAs and iTFAs have similar detrimental effects on health, or whether some bacterial transformations or cattle diet manipulations can be used to improve or mitigate the adverse effects of TFAs, which are highly debated and need to be addressed.

The present article reviews the literature concerning TFAs, including metabolic studies and epidemiological investigations, with the aim of providing an overview of the different health effects of TFAs and identifying if there is a relation between TFA source and its health effects. It gives an account of recent advances in the field and draws comparisons between TFAs from industrial and ruminant origins, putting special emphasis on the latter, as it has been a central topic of debate. Moreover, the review presents and discusses the recent regulatory actions towards elimination of TFA in human diet.

To cover the literature on the topic, multiple databases were included in the research stage of the study. PubMed, Medline, Cochrane Library, Google Scholar, Scopus, and Web of Science were used to find articles relevant to the topic. During the searching process, the key words “trans fatty acid”, “ruminant trans fatty acids”, “industrial trans fatty acids”, “conjugated linoleic acid”, “conjugated linolenic acid”, “conjugated fatty acids”, and “vaccenic acid” were used and each was combined in the search with the keywords “microbiota”, “microbiome”, “LDL and HDL cholesterol”, “cardiovascular disease”, “cancer”, “inflammation”, “insulin resistance”, “diabetes” “oxidative stress”, “atherosclerosis”, “health”, “obesity”, “cardiometabolic disease”, “hypercholesterolemia”, and “trans fatty acid regulation”. In addition to the database search, the reference lists of the articles found were scanned in order not to miss any information on the health effects of different types of TFAs, as well as information on their intake.

Afterwards, the database was filtered to exclude articles that did not include information on health effects. While analyzing the studies, special focus was given to identifying the relationship between TFAs’ origin/type and its adverse/beneficial health effects. Moreover, additional literature was analyzed regarding trans fat regulation and comparisons were drawn between different geographical regions.

The information from the analyzed database for references is presented in a PRISMA diagram (Figure 2).

## 3. Health Effects of TFAs

### 3.1. Obesity

Obesity is recognized as a chronic or noncommunicable disease and an important public health problem due to its comorbidities, such as dyslipidemia, hypertension, and insulin resistance, that frequently occur in this condition. It is known that a diet with a high fat content can lead to obesity, especially in diets rich in saturated fatty acids. Contrariwise, the TFAs’ effects on obesity induction is controversial and discussed below.

Effects on body fat mass (BFM) reduction by CLA were found in the literature [28,29,30,31]. In particular, Gaullier and collaborators reported that a CLA mixture (80% trans-10, cis-12 and cis-9, trans-11 isomers) significantly reduces BFM among healthy overweight adults [30]. Blankson and coworkers reported that 3.4 g/day of CLA (50:50 ratio of cis-9, trans-11 and trans-10 cis-12 isomers) given to overweight and obese humans for 12 weeks significantly reduces BFM, but not body mass index [31]. Another similar study including the same mixture of CLA isomers was conducted among overweight and grade I obese Chinese subjects. CLA (50:50 ratio of cis-9, trans-11 and trans-10, cis-12 isomers) supplementation of 3.4 g/day for 12 weeks resulted in a decrease in body weight, body mass index, body fat mass, fat percentage, subcutaneous fat mass, body fat percentage, and waist-to-hip ratio. However, there was no change in lean body mass after CLA supplementation for 12 weeks [32]. Based on another study of 122 obese subjects without comorbidities, after 1 year of CLA supplementation, no significant difference in body weight or body fat regain was observed. Subjects were exposed to a 3.4 g/day CLA supplementation (consisting of 39% cis-9, trans-11 isomers and 41% trans-10, cis-12 isomer as triacylglycerol; the remaining 20% of the lipid content consisted of triacylglycerol). Similar health effects were also found in relation to CLA-precursor vaccenic acid, even if mostly in animal model studies. In the first study of Wang and colleagues, the 3 week 1.5% vaccenic acid supplementation in obese and insulin-resistant JCR:LA-cp rats determined the 40% decrease in fasting triglyceride concentrations [33]. A later study on the same animal model supplemented with 1% vaccenic acid for 8 weeks showed a stimulated adipose tissue redistribution, a reduced total body fat, and a decreased adipocyte size in comparison to control without vaccenic acid supplementation, thus alleviating the features of metabolic syndrome [34]. However, similar results were not found in another obesity animal model, the fa/fa Zucker rat. Indeed, although the authors found a reduced adipocyte size after a 1.5% vaccenic acid supplementation for 8 weeks, the other obesity-related metabolic abnormalities neither worsen nor were mitigated [35]. In another double-blinded, crossover trial 27 individuals with overweight, borderline hypercholesterolemia were exposed to CLA supplementation. Participants consumed 3.5 g/day of sunflower oil (50:50 ratio of trans-10, cis-12, and cis-9, trans-11 isomers, Clarinol G-80, and cis-9, trans-11 isomer). The CLA intervention did not affect changes in body weight, body composition, or blood lipids [36].

Different data were found in relation to iTFAs, where a positive correlation between iTFA consumption and obesity was found. In the EPIC-PANACEA (Physical Activity, Nutrition, Alcohol, Cessation of Smoking, Eating out of Home, and Obesity) study, the doubling of elaidic acid was associated with a decreased risk of weight loss [37]. In the Spanish INMA (INfancia y Medio Ambiente) study similar results were found in 4- to 5-year-old children. Indeed, iTFA intake of >0.7 g/day was positively associated with being overweight, including obesity; differently, in the same study, no significant association for nTFAs was found [38]. The same results were confirmed in the cross-sectional study of Honicky and colleagues on children and adolescents who underwent a procedure to treat congenital heart disease. In this study, the patients that exceeded the iTFA intake recommendation of 1% of energy had a 5-fold increase of central adiposity [39]. Some recent explanations have related the iTFA consumption with the increased genetic susceptibility of the obesity-associated gene polymorphisms (rs1121980, rs1421085, and rs8050136) and BMI or weight changes, highlighting the important role that iTFAs can have on human metabolism [40].

### 3.2. Insulin Resistance

Type 2 diabetes is the most common hyperglycemic pathology caused by the condition in which the body becomes resistant to insulin. As seen in obesity, the consumption of a high fat diet and a high intake of saturated fatty acids are associated with an increased risk of type 2 diabetes. Similarly, some evidence shows that a high intake of TFAs can result in increased insulin resistance [41]. Clinical trials including healthy and overweight or diabetic individuals showed that TFAs worsen insulin resistance in overweight or diabetic individuals (those with preexisting insulin resistance) but may have smaller effects in young lean subjects. Those with preexisting insulin resistance had both higher fasting insulin and insulin resistance in the case of palm oil and partially hydrogenated soybean oil diets, compared with canola or soybean oil diets [42]. In the paper of Itcho and colleagues [43], two Japanese cohorts of Japanese Americans and native Japanese were compared in relation to serum elaidic concentration and its association with insulin resistance. The serum elaidic acid concentration was significantly higher in the Japanese Americans than in the native group. In the native Japanese, the serum elaidic acid concentration of the group with diabetes mellitus was significantly higher in comparison to normal glucose tolerance and impaired glucose tolerance groups (16 µmol/L, 10.8 µmol/L, and 11.7 µmol/L, respectively) [43]. Similar positive associations were found in a US cohort of 3801 participants where total TFAs, elaidic acid, palmitelaidic acid, and vaccenic acid were significantly positively associated with fasting glucose, fasting insulin, insulin resistance index, and glycated hemoglobin >6.5% [44]. A study including 183 individuals (127 individuals without insulin resistance and 56 with insulin resistance) indicated that trans fat intake impairs insulin sensitivity affecting insulin signaling via intracellular kinases, which alters insulin receptor substrates [45]. Some studies tried to explain the molecular mechanism behind the impaired glucose tolerance induced by TFAs, and some correlations have been found with impaired insulin-dependent GLUT4 translocation [46], impaired IRS1-PI3-Akt pathway involved in GLUT4 translocation [47], and ER-stress and inhibition of insulin receptors [48]. Conflicting results were instead found for vaccenic acid. Indeed, despite a study that compared the fatty acid composition in triglycerides between normoglycemic-normoinsulinemic and hyperglycemic-hyperinsulinemic men showing vaccenic acid was positively correlated with the second group [49], many other lines of evidence suggest that vaccenic acid can have a positive impact on insulin resistance. Recently, high plasmatic levels of the CLA-precursor vaccenic acid were found to be inversely associated with insulin resistance. In particular, it was related to 17%, 32%, and 39% lower risks of incidence of type 2 diabetes in black, Hispanic, and Chinese American populations. Conversely, palmitoleic acid was found to be positively associated with a greater risk (21%) of incidence of type 2 diabetes [50]. Moreover, CLA supplementation did not result in significant differences in reported adverse effects or indexes of insulin resistance; however, it has increased the number of leukocytes [51]. Another double-blind, randomized, controlled trial study was conducted including 62 healthy subjects who received either 3.9 g/day CLA or 3.9 g high oleic acid sunflower oil for 12 weeks. The CLA capsule contained 65.9% CLA (consisting of 29.7% cis-9, trans-11 isomer and 30.9% cis-10, trans-12 isomer), and minimal amounts of other CLA isomers (2.9%). In addition, the capsule contained oleic acid (18:1*n*-9, 24.7%), and a small amount of palmitic acid (16:0; 3.5%), stearic acid (18:0; 1.3%), and linoleic acid (18:2*n*; 9; 12; 1.9%). CLA supplementation reduced plasma insulin concentrations in response to an oral glucose challenge in healthy women, but there were no CLA-specific effects on body composition, energy expenditure, or appetite [52,53].

### 3.3. Cardiovascular Disease (CVD)

The World Health Organization (WHO) identified CVDs as the most common cause of worldwide death. CVDs included a group of disorders of the heart and blood vessels such as coronary heart disease, cerebrovascular disease, deep vein thrombosis, pulmonary embolism, peripheral arterial disease, rheumatic diseases, and congenital heart diseases. They are usually associated with atherosclerosis, caused by fatty deposits inside the arteries, and determining the hyperlipidemia as a risk factor. There are numerous studies providing evidence that iTFAs increase the risk of coronary heart disease [54]. It has been stated that TFA consumption perturbs the body’s ability to metabolize essential fatty acids (including omega-3 fatty acids) leading to changes in the phospholipid fatty acid composition in the aorta, thus increasing the CVD risk [55]. Based on a study including patients with CVD, it has been proved that changes in the composition of phospholipids and fatty acids of membrane lipids largely determine the development of CVD [56]. Other studies have supported that some plasma phospholipid SFAs, such as palmitic (16:0) and stearic (18:0) acids increase the risk of coronary heart disease [57,58]. The Nurses’ Health Study, which is the largest epidemiological study in this field, showed a significant, positive association between the intake of iTFAs and heart disease risk [22]. In the Framingham study, a significant, moderately increased risk of heart disease was found associated with the intake of margarine, which is a major source of TFAs [59]. Moreover, according to the Finnish Alpha-Tocopherol, Beta-Carotene Cancer Prevention Study, the increased risk of cardiac death is positively associated with the intake of total TFAs as well as elaidic acid and TFAs from hydrogenated vegetable fats [60]. In addition, Mozaffarian et al. [42] found that the consumption of iTFAs has adverse effects on plasma lipids (for example, higher LDL cholesterol, lower HDL cholesterol, and higher total/HDL cholesterol ratio), proinflammatory effects (for example, higher levels of tumor necrosis factor-α activity, higher interleukin-6, and higher C-reactive protein), and endothelial dysfunction. Harvey et al. [61] found that linoelaidic acid appeared to be potentially more detrimental than elaidic acid. Li et al. [62] showed that linoelaidic acid induces a stronger lesion effect on HUVECs compared to elaidic acid, but further studies will be necessary to identify the underlying mechanism involved in these processes. Interestingly, evidence was brought that TFAs can increase the risk of CVDs to an extent equal to or even greater than SFAs [63,64]. It is noteworthy that according to the WHO report, about 540,000 deaths yearly can be attributed to the intake of TFAs from industrial sources [65].

It should, however, be noted that the intake of TFAs from ruminant sources has either not been associated or has been negatively associated with the risk of coronary heart disease [66]. Indeed, although an early study by the US Department of Agriculture showed that vaccenic acid raises both HDL and LDL cholesterol [67] and this was further supported by the study of Øie and colleagues [68] that found high levels of vaccenic acid were associated with disease severity and mortality in 183 patients with chronic heart failure [68], subsequent studies disconfirmed this. According to the Nurses’ Health Study, there is a nonsignificant, inverse association between the intake of nTFAs from ruminant sources and the risk of heart disease, indicating the association between heart disease and TFAs was due to partially hydrogenated vegetable fat rather than to isomers from ruminant sources [69]. The same pattern was observed in the Alpha-Tocopherol, Beta-Carotene Cancer Prevention Study [70]. Additionally, studies have shown that vaccenic acid can be beneficial against CVD. Indeed, Bassett and coworkers [71] found that butter enriched in vaccenic acid did not induce atherosclerotic plaque formation and reduced the serum cholesterol and triglyceride levels in LDL receptor deficient mice in comparison to regular butter [71]. Later, Kim and colleagues [72] tried to explain the mechanism behind the anti-atherogenic effect of vaccenic acid and in their in vitro study found that, unlike elaidic acid, vaccenic acid and CLA did not induce vascular smooth muscle cell proliferation and migration, steps that normally take place during atherosclerotic plaque formation [72]. Furthermore, some anti-inflammatory activities were found in relation to CLA and vaccenic acid. Initially these activities were only attributed to cis-9, trans-11 CLA, which was found to modulate the transcription of TNF-alpha, IL-12, and IL-16, and the production of IL-12 [73]. Finally, based on the meta-analysis of Derakhshande-Rishehri and coworkers [74], both CLA supplements and foods enriched with CLA caused a significant reduction in LDL cholesterol levels among the healthy adult population. Foods enriched in CLA, in comparison with CLA supplementation, had a beneficial effect on the whole lipid profile, although only the effect on LDL cholesterol levels was statistically significant [74].

### 3.4. Cancer

Cancer is the second leading cause of worldwide death, accounting for an estimated 9.6 million deaths. Lung, prostate, colorectal, stomach, and liver cancer are the most common types in men. Breast, colorectal, lung, cervical, and thyroid cancer are the most common in women. Cancer starts when cells grow out of control and spread throughout the body. According to Stender et al. [22], there is a positive relationship between TFA intake and the incidence of breast and large intestine cancer [22]. Similarly, more recently, Ardisson Korat and colleagues [75] found a positive relationship between TFA levels in red blood cell membrane and the risk of large B cell lymphoma, especially for elaidic and vaccenic acid [75]. In the European Prospective Investigation into Cancer and Nutrition (EPIC), Yammine et al. [76] suggested a positive association between ovarian cancer and intake of industrial trans elaidic acid. In the same EPIC study the dietary iTFAs, particularly elaidic acid, were positively associated with rectal cancer [77]. TFAs have been hypothesized to influence breast cancer risk; however, relatively few prospective studies have examined this relationship. Matta et al. [78] support the hypothesis that higher dietary intakes of iTFAs, in particular elaidic acid, are associated with elevated breast cancer risk. The effects of elaidic acid on colon carcinogenesis remain controversial; plasma concentration of elaidic acid is higher in colon adenoma patients than in healthy controls [79]. Ohmori et al. [80] showed that elaidic acid might provide metastatic potential to colorectal cancer cells. It is interesting that in the Netherlands Cohort Study on Diet and Cancer, a weak positive relationship between CLA intake and the incidence of breast cancer was found (for CLA, a total of cis-9, trans-11 and trans-9, cis-11 isomers were used) [81]. Moreover, according to Benjamin and coworkers [82], the negative effects of CLAs can also include colon carcinogenesis induction.

Nevertheless, in the case of uncontrolled cell proliferation, the TFAs’ action is elusive in the literature data. In addition, the role of vaccenic acid is controversial, being deemed both dependent and independent from its conversion to CLA. Indeed, in vivo animal studies in rats showed that the anticarcinogenic actions of vaccenic acid against mammary carcinogenesis is dependent on its conversion to cis-9, trans-11 CLA by Δ9-desaturase [83,84,85]. In an early in vitro study, it was stated that vaccenic acid showed no antiproliferative effects in comparison to cis-9, trans-11 CLA that determined cell differentiation in Caco-2 cells [86]. Later studies confirmed the anticarcinogenic action of vaccenic acid independently from CLA isomers in breast carcinoma cells [87,88] finding correlations with the mitochondrial-mediated apoptosis pathway [89]. Vaccenic acid inhibits proliferation and induces apoptosis of human nasopharyngeal carcinoma cells [89]. The consumption of CLA from dairy products reduces breast and colorectal cancer. A 3 year study on more than 60,000 women suggests that high intake of high fat dairy foods and CLA may reduce the risk of colorectal cancer, the third most common incident cancer worldwide [90]. Another study conducted on more than 500,000 individuals (4992 incident cases of colorectal cancer) from America and the US suggests that higher consumption of milk and calcium is inversely associated with the risk of colorectal cancer [91]. Mohammadzadeh and coworkers [92] reported that based on a randomized, double-blind study, including 34 patients with rectal cancer, who have received 3 g/day CLA for 6 weeks, CLA supplementation improved inflammatory factors, matrix metalloproteinase-2, and matrix metalloproteinase-9 as biomarkers of angiogenesis and tumor invasion [92]. Several other cohort studies also suggest that increased consumption of total dairy products and CLA-rich foods reduces the risk of breast cancer [93,94]. However, it is important to note that the literature regarding CLA effects on cancer varies from study to study. As mentioned above, according to other studies, CLAs may lead to colon and breast cancer [81,82]. It has also been suggested that the consumption of dairy products (particularly cheese) may be a factor against the development of breast cancer, supposedly due to CLA alone or together with other ingredients. In contrast, high consumption of processed ruminant meat, reflected in an increased proportion of arachidonic acid in serum fatty acids, may increase the risk of breast cancer. It shall also be highlighted that when studying the effects of a CLA-enriched diet on cancer, it is often impossible to assess the independent effects of CLA [85]; therefore, it can be concluded that there is insufficient evidence to determine whether CLA ingestion has a significant effect on cancer [95], and its mechanisms of action should be studied more thoroughly.

### 3.5. Inflammation

In recent years, systemic chronic inflammation was linked to several diseases that determine the leading causes of worldwide death, including CVDs, cancer, diabetes mellitus, chronic kidney disease, nonalcoholic fatty liver disease, and autoimmune and neurodegenerative disorders. It has been proved that TFA consumption influences multiple risk factors, including increased systemic inflammation [96,97]. Studies in humans usually show a relation between iTFAs and higher levels of inflammatory markers. More precisely, based on a controlled feeding study conducted among men, consumption of TFAs, stearic acid, and LMP (the sum of lauric (L), myristic (M), and palmitic (P) acids) diets increased the concentrations of inflammatory markers, such as C-reactive protein (CRP), fibrinogen, and interleukin 6. Moreover, stearic acid showed to increase the risk of cardiovascular disease through an increase in fibrinogen concentrations [8]. Another study conducted among ambulatory patients with chronic heart failure showed that, in contrast to palmitoleic acid, trans isomers of oleic and linoleic acids were associated with higher inflammatory marker concentrations [28]. In a similar study including obese humans without comorbidities, CLA (50:50 ratio of cis-9, trans-11 and trans-10, cis-12 isomers) supplementation of 6.4 g/day for 12 weeks increased lean body mass, as well as markers of inflammation, such as C-reactive protein and interleukin-6. However, CLA supplementation did not change body fat mass, weight, body mass index, resting energy expenditure, or respiratory quotient [98]. A different in vitro study confirmed the potential negative effect of TFAs on inflammatory parameters. Indeed, the authors showed that elaidic acid, linoelaidic acid, and vaccenic acid, but not their corresponding cis isomers, enhanced the extracellular ATP-induced apoptosis activating the ASK1-p38 pathway leading to inflammation and cell death typical of TFA-related atherosclerosis [99]. Similarly, the National Health and Nutrition Examination Survey (NHANES) conducted on 5446 participants from 1999 to 2010 and measuring plasmatic TFAs and the markers of dietary inflammation revealed that elaidic acid, linoelaidic acid, and vaccenic acid were positively associated with these markers, highlighting their role in the CVD’s initiation and progression [100]. Higher consumption of TFAs is a risk factor for several inflammatory diseases including inflammatory bowel disease. Higashimura et al. [101] demonstrated that elaidate, a trans-isomer of oleate, enhanced the induction of IL1β in colonic mucosa of colitis model mice, and increased colonic damage and myeloperoxidase activity. A study using a mouse model of hyperlipidemia demonstrated that elaidate significantly increased ROS production and NADPH oxidase expression in the aortic vessel wall [102]. TFAs have been reported to promote vascular diseases by promoting apoptosis and inflammation of vascular endothelial cells. Hu et al. [103] investigated the effects of elaidic acid and vaccenic acid on human umbilical vein endothelial cell (HUVEC) function. The results showed that both elaidic acid and vaccenic acid reduced HUVEC viability, induced cell membrane damage, and increased membrane permeability. However, the damage by elaidic acid was significantly stronger than that by vaccenic acid. The expression of ICAM-1, VCAM-1, and IL-6 and the PGE secretion were increased and the effect of elaidic acid was significantly higher than that of vaccenic acid [103].

Contrary to the aforementioned findings, some studies have reported that TFAs do not increase markers of inflammation [104,105]. A random study including 61 healthy adults revealed that a high intake of iTFAs and conjugated linoleic acid did not substantially affect plasma concentrations of inflammatory markers [104]. A recent randomized, crossover study in 106 healthy adults suggested that there is no effect of TFAs, regardless of the source, on inflammation and cell adhesion. Neither vaccenic acid nor iTFAs had an impact on inflammation and cell adhesion [105]. Moreover, subsequently some anti-inflammatory activities were isolated and found for vaccenic acid. Indeed, Lee and colleagues [106] found that physiological concentrations of vaccenic acid enhanced the IL-10 production and suppressed TNF-alpha in mice after LPS treatment [106]; in addition, vaccenic acid and palmitoleic acid were found to reduce TNF, VCAM-1, and SOD2 inflammatory gene expression in umbilical vein endothelial cells and human hepatocellular carcinoma cells [107]. Similarly, Da Silva and coworkers found that vaccenic acid and palmitoleic decreased inflammatory prostaglandins excretion induced by TNF-alpha in vascular endothelial cells, but simultaneously increased the F2-isoprostanes oxidative stress markers [108]. Nevertheless, still nowadays the evidence in the literature is controversial as to whether the anti-inflammatory action of vaccenic acid can be attributed to itself or to its endogenous partial conversion to cis-9, trans-11 CLA. Indeed, Jaudszus and colleagues, through the analysis of the fatty acids profile before and after vaccenic acid treatment, showed that the cis-9, trans-11 CLA amount remain unaltered, suggesting that the anti-inflammatory action of vaccenic acid was independent from cis-9, trans-11 CLA [109]. However, more recently Li and colleagues [110] showed that when the conversion of vaccenic acid in cis-9, trans-11 CLA was inhibited by leptin, an upregulation of inflammatory ICAM-1, VCAM-1, and IL-6 occurred; conversely without leptin the vaccenic acid bioconversion rate in cis-9, trans-11 CLA was 23%, and a remarkable down-regulation of ICAM-1, VCAM-1, and IL-6 occurred [110]. Finally, recent epidemiological work cross-sectionally associated circulating levels of trans vaccenic acid in humans with lower adiposity, diabetes risk, and systemic inflammation [111].

## 4. Do the Health Effects of TFAs Differ Depending on Their Origin?

Our above analysis shows that, overall, iTFAs are shown to be more deleterious than ruminant-origin TFAs; however, the latter are also considered to have some adverse health effects in certain cases.

More recent inquiry (independent of the dairy industry) has found in a 2003 Dutch meta-analysis that all TFAs, regardless of natural or artificial origin, equally raise LDL cholesterol and lower HDL cholesterol levels [22]. According to Gebauer and colleagues [105], TFA (industrial ruminants, and CLA) raises the ratio of LDL to HDL cholesterol, which in turn increases the risk of coronary heart disease. However, ruminant TFAs and CLA have less effect on the LDL to HDL cholesterol ratio than iTFAs, although further studies will be needed to decide whether this difference is real or due to chance [19]. Verneque et al. [112] reported that at levels up to 1.5–7% of energy, the effect of rTFA seemed to be more than iTFA. However, rTFA seemed to be less harmful than iTFA for HDL cholesterol, although for total cholesterol and LDL cholesterol it might be worse [112].

The US National Dairy Council has asserted that the TFAs present in animal foods are of a different type than those in partially hydrogenated oils, and do not appear to exhibit the same negative effects. CLA has various beneficial physiological effects that are associated with reduced cardiovascular heart disease [113]. These effects include changes in body composition [114] and lower insulin resistance. There are other health effects, such as anticarcinogenic, antiatherogenic, anti-obesity, and antidiabetic effects and immune system enhancement. However, the latter have been investigated mainly in different animal models, and not in humans [115]. Mazidi and colleagues [116] found a negative association between TFAs and telomere length in US adults, where shorter telomere length is associated with a broad range of age-related diseases. Nevertheless, only palmitelaidic and linoelaidic were significant and negatively associated with telomere length [116]. Finally, more recently a study was conducted to ascertain the relationship between plasma TFAs and overall cause of mortality. This study showed that the iTFA elaidic acid might be associated with mortality. Regardless, no nTFAs were inversely associated with the risk of mortality [117]. Thus, it can be concluded that the source could be important for the health influence of TFAs. While iTFAs are known for many deleterious health effects, nTFAs are shown to be less harmful and, in some cases, beneficial. However, more data are needed to draw more precise conclusions. Oteng and Kersten [27] reported that there is an ongoing debate over whether iTFAs and nTFAs exert the same effects on cardiovascular health. Evidence from both preclinical and clinical studies indicated the two types of TFAs can behave similarly or differentially, depending on the biological pathway and clinical parameters under investigation [27]. According to Verneque et al. [112], both sources of TFA can increase cardiometabolic risk parameters, especially lipid profile. However, the dose of TFA and the whole composition of the food must be considered [112].

It is important to understand how TFAs are formed and whether there is a method to reduce the health effects of deleterious TFAs and to bolster those of beneficial ones. Finally, considering the importance that the study of gut microbiota has acquired in recent years to explain biochemical pathways as the basis of the etiology of various pathologies, the TFA–gut microbiota interactions study appears to be fundamental to our understanding of the TFAs’ dualism in relation to health.

## 5. nTFAs and iTFAs’ Impact on Gut Microbiota: Eubiotics or Xenobiotics?

In an attempt to understand the dual role that TFAs have on health, the analysis of their impact on gut microbiota could be valuable. Indeed, in recent years, many diseases have been related to gut microbiota functions. Considering that diet undoubtedly influences the composition of gut microbiota, providing nutrients for both the host and the bacteria, it is increasingly important to analyze the action of each molecule introduced to foods on gut microbiota health. Most is known in relation to the healthy systemic effects of nTFAs, but what is their impact on human gut microbiota eubiosis? Are they something similar to xenobiotics, or do human gut microorganisms have the same ability as ruminal microbiota to convert them into linoleic acid? The first evidence that CLAs may also be formed during the metabolism of linoleic acid by human gut microbiota came from the ex vivo study of Howard and Henderson [118], where linoleic acid and alpha-linolenic acid were incubated with human fecal suspension.

Few later works investigated the effect of dietary CLAs’ intake in vivo on the microbiota of humans, but mostly in animal models and with mixed results. In an early study on moderately hyperlipidemic and overweight volunteers who consumed three types of milk with different types and levels of CLAs for 56 days, only a significant decrease in *Bifidobacteria* numbers in all three groups was shown, indicating that the effect is more attributable to milk supplementation per se, rather than CLA content [119]. In a later paper, the change in microbiota composition was analyzed in obese mice fed with a high fat, high sucrose diet alone or supplemented with 1% of cis-9, trans-11 CLA or trans-10, cis-12 CLA. After 8 weeks of feeding, faecal alpha diversity and weight decreased in all groups. However, a trans-10, cis-12 CLA enriched diet determined the increase of *Butyrivibrio, Roseburia, Lactobacillus, Actinobacteria,* and a new identified strain named *Ileibacterium valens* of the *Allobaculum genus*, with a reduction in *Prevotella* e *Clostridium* genera. Conversely, cis-9, trans-11 CLA supplemented diet mice were the most enriched in *Bifidobacteria* [120]. Nevertheless, in the recent paper of Li and colleagues, the supplementation of 0.2% CLA mixture (50% cis-9 trans-11 CLA, and 50% trans-10, cis-12 CLA) in C57BL/6J mice fed a high fat diet to induce obesity showed sex-dependent alteration of gut microbiota compositions. For example, *Desulfovibrio*, a genus that produces hydrogen sulfide, was reduced in male mice but enhanced in female mice. Moreover, the pro-inflammatory *Ruminococcus* and *Lachnospiraceae*, positively linked to type 2 diabetes, increased only in female mice. Finally, a high fat diet increased the circulating LPS in both genders; the CLA treatment decreased this level only in male mice and it aggravated these conditions in female mice [121]. In the key research of Marques and colleagues, the pros and cons of 8 weeks’ trans-10, cis-12 CLA supplementation in mice were evaluated and correlated with gut microbiota composition and short-chain fatty acid production. Decreased visceral fat mass and higher concentrations of acetate, propionate, and isobutyrate, as well as short-chain fatty acids, were found and correlated with changes in microbiota composition, such as lower *Firmicutes* and higher *Bacteroidetes* proportions. However, among *Bacteroidetes*, a high content of *Porphyromonadaceae* was found and previously correlated with the induction of hepatic steatosis and altered lipid metabolism [122]. Moreover, a decreased abundance of the enteric bacteria *Lachnospiraceae* and *Desulfovibrionaceae* were found, with the former having documented positive effects against *Clostridium difficile* colonisation in mice [123], and lower abundance was correlated with cirrhosis [124]. Meanwhile the *Desulfovibrionaceae* family was associated with impaired glucose tolerance and metabolic syndrome [125]. Finally, in another important paper [126], the effects of CLA (50:50 mixture of cis-9, trans-11 CLA and trans-10, cis-12 CLA) at different concentrations in the range of 2.5–40 mg/day were analyzed in C57BL6/J mice as preventive and curative treatments against dextran sulphate sodium (DSS)-induced colitis (7 days of CLA and subsequently 7 days CLA + DSS). The results showed that CLA isomers at high concentrations (40 and 20 mg/day) are able to alleviate colitis symptoms and to reduce DSS-induced inflammation, modulating oxidative stress enzymes and inflammatory cytokines, and increasing mucin-2, goblet cells, and tight junction proteins. Furthermore, the 40 mg/day CLA treatment was able to rebalance the gut dysbiosis induced by DSS, reducing the *Bacteroides* (negatively associated with IL-10 and tight junction proteins, thus having negative effects on colitis) and increasing *Bifidobacterium* and *Odoribacter*, known as a producer of short-chain fatty acids, as butyrate, with anti-inflammatory abilities [126].

If the debate is still open for the effect of nTFAs on human microbiota, the knowledge about the relation of iTFAs and human microbiota is still in its infancy. Indeed, only five papers were published with conflicting results and only in animal models. In the paper of Ge and colleagues [127], soybean oil and partially hydrogenated soybean oil in low (4.27%) and high (23.60%) concentrations were administered in mice for 8 weeks. The authors stated that a clear dysbiosis of the gut cecum microbiota was found, with an enhancement in the harmful bacteria abundance, such as *Proteobacteria,* the second most abundant phylum after *Firmicutes*, instead of *Bacteroidetes* as in the control group, and *Desulfovibrionaceae* [127]. This last taxon is known to increase the permeability of the gut mucus membrane due to excessive H_2_S production, and therefore increases inflammation of the colonic epithelium [128], correlated in the present paper with high expression of inflammatory cytokines [127]. In support of the iTFA-induced dysbiosis, in the same study, a relative abundance decrease in beneficial bacteria was recorded for *Bacteroidetes, Lachospiraceae, Rikenellaceae,* and *Bacteroidales S24-7.* A decreasing abundance of the beneficial *Lachospiraceae* was also found after administration of nTFAs discussed above [122], and, thus, it is likely that TFAs in general act as toxic elements for this bacteria taxon. Reduced abundance of *Rikenellaceae* was previously correlated with nonalcoholic fatty liver disease [129], and in this study impaired liver functions were found and probably relate to this taxon dysbiosis [126]. Beyond expectations, iTFA supplementation also enhances the relative abundance of *Lactobacillaceae*, considered beneficial bacteria, enough to also be used as a probiotic [127]. However, the increase of *Lactobacillaceae* cannot be correlated with a healthy status considering that in previous studies its increase was related to impaired conditions in mice, for example, obesity and steatohepatitis [130]. Similarly, the same authors in their recent paper confirm this dysbiotic effect of iTFAs’ supplementation in the small intestine of C57BL6/J mice. Indeed, after 8 weeks of feeding with soybean oil and partially hydrogenated soybean oil in low (10% kcal from fat) and high (45% kcal from fat) concentrations, *Proteobacteria*, *Lactobacillus*, and *Desulfovibrio* were found to be significantly higher in the iTFA groups in comparison to non-iTFA groups [131]. Conversely to what was shown above, in the study of Carvahlo and coworkers [132], the authors stated that high fat and iTFAs, in the form of partially hydrogenated vegetable oils, are less harmful than expected [132]. The research contemplated a 13 week13-week experimental period in which four different high fat diets were administered to mice, and in which the alternation of partially hydrogenated fats and whey protein hydrolysates (with demonstrated anti-inflammatory properties) were combined and compared with a control diet. In the discussed results, the authors stated that, even though *Bacteroidetes* decreases and *Firmicutes* increases, the general predominance of *Bacteroidetes* was maintained. Conversely to the previously described studies, here the *Proteobacteria* abundance was higher in an unhydrogenated oil diet (composed at 93% of soybean oil), particularly when not associated with whey protein hydrolysates. Thus, the iTFA diet seemed to be more conservative for the original microbiota population in comparison to the unhydrogenated high fat diet. However, some changes were recorded, in particular an increase in *Parabacteroides goldsteinii* and *Akkermansia muciniphila* in an iTFA-containing diet, whereas *Butyricimonas*, *Actinocorallia*, and *Natronincola* decrease [132]. *Parabacteroides goldsteinii* is a pathogenic species associated with bacteraemia in humans [133] and it increased in this study after iTFA supplementation. On the contrary, *Butyricimonas synergistica* and *B. virosa* decrease and were associated with bacteraemia in humans [134], while *Akkermansia muciniphila* increased and was correlated with the improvement of several metabolic parameters [135]. From this mixed result, the authors stated that iTFA supplementation mostly does not determine the dysbiosis of gut microbiota in mice [132]. Nevertheless, two recent papers further support the dysbiotic effect of the iTFAs. In the paper of Hua and coworkers [136], Sprague Dawley rats fed 1% and 8% of partially hydrogenated soybean oil in a high fat diet showed the same altered parameters as the animals fed a high fat diet with a high level of lard. Indeed, the diets significantly induced obesity and changes in the microbiota composition as shown by the inversely altered *Firmicutes* and *Bacteroidetes* ratio, the increase in *Proteobacteria* and *Bacteroides*, and the decrease in *Muribaculaceae*, that may affect inflammation [136]. Similarly, in the paper of Okamura and colleagues [137], C57BL6/J mice fed a high trans, high sucrose diet, in addition to an increase of elaidic acid in the liver and serum together with increased blood glucose levels and intestinal inflammation, showed a significantly higher abundance of the family *Desulfovibionaceae*, belonging to the phylum *Proteobacteria* [137].

In conclusion, TFAs’ influence on gut microbiota seems to follow the same trend previously seen for other health conditions: nTFAs are more beneficial for health in comparison to iTFAs, which are more similar to xenobiotics. However, considering the mixed results obtained on both fronts, and the shortage of human investigations, further and deeper analyses are needed to outline the real TFA effects on human gut microbiota.

## 6. Comparison of the Regulations and Reduction Policies of TFAs Worldwide

As it has been noted above, iTFA intake has a negative impact on the heart. Hence why, in recent years, different health organizations have recommended the lowering of iTFA intake as much as possible [138].

According to the European CVD statistics, trans fat consumption significantly increases the risk of CVD, which is the main cause of death for Europeans under the age of 65 [139]. The WHO reports that a 2% increase in energy from trans fats leads to a 25% increase in the risk of death from CVD [140].

According to the National Academy of Sciences of the USA, dietary TFAs are more deleterious for coronary artery disease than saturated fatty acids [138]. Interestingly, EFSA stated that even though the effect of TFAs on heart health may be greater than that of saturated fats, given the current intake levels of TFAs, their potential to increase cardiovascular risk is much lower than that of saturated fats [141]. A comprehensive report was published by the National Academy of Science, Institute of Medicine in July 2002, recommending that the intake of TFAs should be as low as possible, without differentiating between iTFAs and TFAs from ruminants [142]. The World Health Organization recommended in 2003 that all trans fats be limited to less than 1% of overall energy intake. EFSA Opinion 2004 [2] suggested that, from 2 April 2021 onwards, foods in the EU intended for consumers must contain less than 2 g of iTFAs per 100 g of fat. The first European country that has imposed a limit on iTFAs in food is Denmark. The limitation to 2 g per 100 g of fat has been quite successful as CVD mortality has decreased significantly in 3 years.

According to the scientific opinion published by EFSA in 2018 [141], the majority of European countries still do not limit the content of trans fats in food. It should be mentioned that the policy option on the TFA ban includes the legislation limiting only the amount of iTFA, but doesn’t apply to TFAs from natural sources (i.e., nTFA). Moreover, the European Commission proposed a legal limit on iTFAs in food according to which, “The content of trans fat, other than trans-fat naturally occurring in animal fat, in foods which is intended for the final consumer, shall not exceed 2 g per 100 g of fat.” This provision came into force in April 2021. The European Margarine Association (IMACE) welcomed the adopted legislation by the European Commission setting a legal limit on the content of iTFAs in food sold in retail and intended for the final consumer.

In 2020, the Pan American Health Organization (PAHO) published the 2020–2025 plan of action for the elimination of iTFAs [143]. Two main policies are mentioned as effective approaches to eliminate iTFAs. The first approach uses legislative or regulatory measures to limit iTFA to no more than 2 g per 100 g (2%) of total fat in all food products, including, but not limited to, fats and oils. Similar to EU practice, the PAHO’s first approach applies to domestic and imported products but excludes TFAs from ruminant sources. The second and more recent policy approach is to ban partially hydrogenated oils (PHO), the major source of dietary iTFA. It is worth mentioning that the US has reclassified PHO as no longer “generally recognized as safe”. Moreover, Canada has listed PHOs among “contaminants and other adulterating substances in food”. Peru and Thailand have adopted measures that are similar to those of Canada and the US [143,144].

The legal limit on trans fats in the Eurasian Economic Union seem to be less stringent. The Customs Union has set “Technical regulations for fat-and-oil products” (TR TS 024/2011) [145], which came into force in Russia in 2018. According to the regulation, the safety parameter for “trans-isomers of fatty acids” has changed from 20% to 2% of the product’s total fat content. Following the new regulation, foods that contain trans fats, such as margarine, special fats, cream vegetable and vegetable fat spreads, cream vegetable and vegetable fat rendered mixtures, milk fat substitutes, cacao butter, and cacao butter equivalents must contain up to 2% of trans-isomers. Moreover, it is mandatory to indicate the ratio of the product’s fat content on the package. The requirements regarding the content of trans-isomers of fatty acids had been changing since 2013 and the process finished on 1 January 2018. This long period gave fat and oil producers time to adapt to the new requirements, which are now effective in all Eurasian Economic Union (EAEU) countries.

It is important to note that the methodology of food control systems incorporates not only the needs of the country, but also cultural aspects and technical infrastructures. The approaches to food safety control in the EAEU and EU are fundamentally different. The EAEU approach is based on the testing of final products in compliance with specific technical regulations or standards. These differences consequently have an impact on the food safety control system as a whole, including the organization of monitoring and control, inspection, testing, labeling, approaches to food quality, and the responsibilities of government, industry, and consumers. The word “risk” in the legislation of the Eurasian Economic Union is not specific to food products. Risk assessment is defined as the key tool of sanitary epidemiological activities. It should be highlighted that in some cases these notions are raising the profile of the concept of internationally adopted approaches; for example, Appendix 9, “Protocol on Technical Regulation within the Eurasian Economic Union”, the Agreement of the Eurasian Economic Union, defines risk as a combination of the probability of harm and the consequences of this harm impacting human life and health (in terms of the sanitary and epidemiological wellbeing of the residents). Therefore, the approach adopted by the EAEU is hazard-based. In comparison, the EU-adopted modern approach is a risk-based food safety control system. The peculiarity of this approach is that the food business operators are responsible for food safety hazard identification, its consequent risk, and the implementation of practices for prevention, minimization, and elimination of hazards. The government is responsible for monitoring and verifying the effectiveness of the risk control measure. In Regulation (EU) No 1169/2011 [146] trans fats are defined as “fatty acids with at least one non-conjugated (namely, interrupted by at least one methylene group) carbon-carbon double bond in the trans configuration”. Meanwhile, the Union Technical Regulation for Fat and Oil Products [145] omits the definition of TFAs in its “Terms and Definition” chapter. On 24 April 2019, the Commission adopted a Commission Regulation Amending Annex III to Regulation (EC) No 1925/2006 of the European Parliament and of the Council [147] regarding trans fats, other than trans fats naturally occurring in fat of animal origin. The maximum limit of trans fats in food intended for the final consumer and for supply to retail is 2 g per 100 g of fat. The novelty of legal changes is an obligation for business-to-business transmission of information on the amount of trans fats in foods when it exceeds the limit of 2% of fat. It is noteworthy that in 2018, several standards for the permissible level of fatty acid trans isomers established by the EAEU Technical Regulation for Fat and Oil Products [145] came into force after the transitional period had elapsed. The content of trans fats in solid, soft, and liquid margarines, milk fat substitutes, and special purpose fats should not exceed 2% of the total fat content in food products (as opposed to the 8–20% safety limit previously). Nevertheless, there is no labeling requirement for products other than fats and oils in the EAEU, which is also a fundamental difference compared to the EU.

The WHO has developed the TFA Country Score Card (Table 1) which shows countries that have best practice TFA policies, less restrictive TFA limits, other complementary measures, or a national policy commitment to eliminate TFA [144].

### Comparison of the Intake and Health Effects of TFAs

It is important to mention that all food products showed iTFA decline except for certain foods such as cookies, biscuits, bakery products, cake mixes, shortenings, and frozen foods that still contain high amounts of TFAs [7]. Taking into consideration the public health concerns due to TFA contents in processed foods, the global agroindustry applied changes in their manufacturing processes. Overall, different strategies, including new processes, treatments, and reformulations such as the modification of the hydrogenation process, interesterification, fractionation, or the combination of several technologies for TFA reduction, have been implemented [148,149]. Interestingly, the processed food industry has an important role in decreasing the TFA content in food by using alternative sources of fat with zero TFA levels. For example, the replacement of frying fats with medium and high stability vegetable oils in the fast food industry resulted in the elimination of TFAs in products fried in the fats [17].

Li et al. [13] stated that it is urgent to measure and monitor TFA intake globally. However, researchers mention that accurate and reliable data on TFA intake are scare, especially in many low and middle income countries. More recent data from different countries (Figure 3) indicate that average daily TFA intakes are below 1% of daily energy (except in the case of Iran, Lebanon, Canada, Puerto Rico, Brazil, and the US); thus, TFA intake has decreased in general [10,148,149]. It should be noted that some population groups are at risk of exceeding the levels recommended by the World Health Organization, 1% of the energy intake [65,150].

In EAEU member countries, exceedingly high contents of TFAs were reported in various foods. A recent study indicates that some of the dishes surveyed in Kyrgyzstan contained up to 170% of the recommended maximum daily intake of trans fats in one serving [139]. Another study, analyzing the TFA content in commonly available products in Kyrgyzstan, found that 54% of foods showed a content of trans fats ≥1% [151]. Unfortunately, there is no sufficient data regarding the TFA intake levels among the population of EAEU countries. Nevertheless, as previously mentioned, the high intake of iTFAs is associated with an increased risk of coronary heart disease [152]; thus, CHD deaths due to TFA intakes are a raw estimate of TFAs’ burden. To compare TFAs’ burden in the EU and the EAEU, data on coronary heart disease death (%) due to TFA intakes for both EU and EAEU countries are presented in the table below (Table 2), based on the calculation from Wang et al.’s study [29]. Overall, this measure is highly variable from country to country; however, the EU average exceeds the EAEU average. Thus, even though EU member countries have more stringent TFA limits than EAEU member countries, the proportion of CHD deaths due to TFAs, on average, is higher for EU countries.

By drawing the comparison between the aforementioned worldwide policies and legislations, as well as data on the intake of TFAs, it is crucial to highlight that, overall, regulations do not apply to the naturally occurring content of TFAs (nTFAs). Moreover, another important concern is the absence of specific data on nTFAs’ consumption across these countries and regions.

## 7. Conclusions

To conclude, many studies report a direct association of iTFA intake with the risk of coronary artery disease (CAD) and other adverse health effects. Meanwhile, in regard to ruminant TFAs, studies report both a positive, negative, and sometimes no association of ruminant TFA intake with adverse health effects. Thus, TFAs from industrial sources are proved to have many adverse health effects; so why do so many companies still use it? It is because trans fats are easy to use, inexpensive to produce, and last a long time. Trans fats give foods a desirable taste and texture. Many restaurants and fast food outlets use trans fats to deep fry foods because oils with trans fats can be used many times in commercial fryers. Several countries (e.g., Denmark, Switzerland, and Canada) and jurisdictions (e.g., California, New York City, Baltimore, and Montgomery County, MD) have reduced or restricted the use of trans fats in food service establishments. Even though some adverse health effects of ruminant origin TFAs, such as CLAs, have also been reported, the negative effects of nTFAs do not transcend those of iTFAs. Therefore, it is recommended that iTFAs be used as little as possible, while the use of the ruminant depends on its type and the organism of the user. Overall, the dietary recommendations should be focused on restriction of TFA intake rather than potential health effects induced by the different types of TFAs.

## Figures and Tables

**Figure 1 foods-10-02452-f001:**
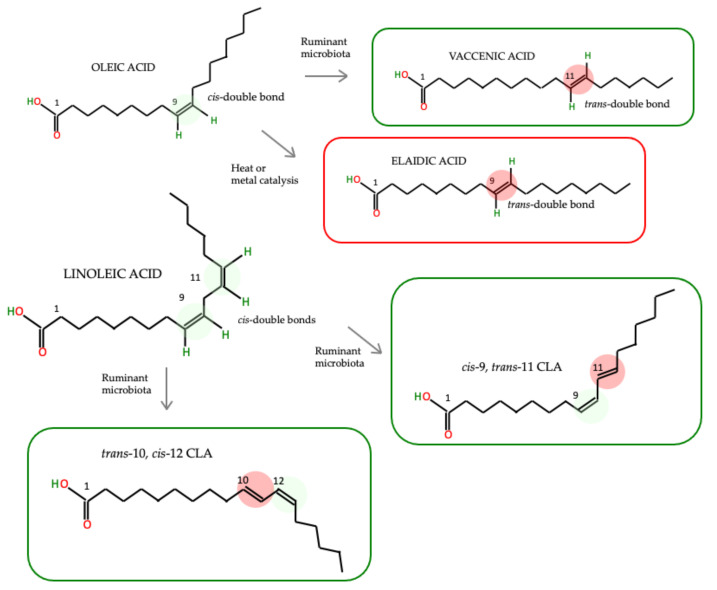
Chemical structure of the major TFA in the green box are nTFA; the red box is iTFA. Green highlighting: cis-double bonds; red highlighting: trans-double bonds.

**Figure 2 foods-10-02452-f002:**
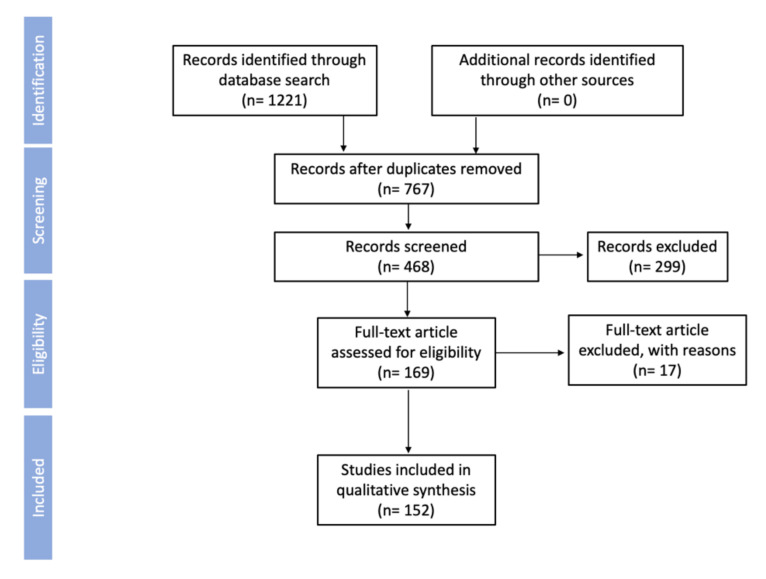
PRISMA flow diagram.

**Figure 3 foods-10-02452-f003:**
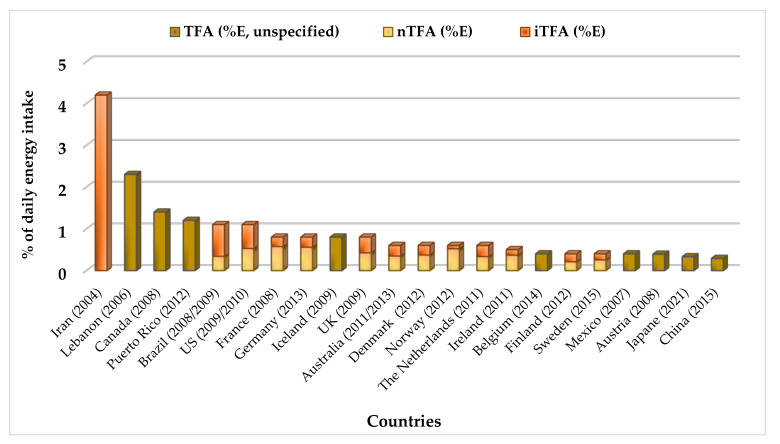
Mean intake of TFA as % of daily energy intake in different countries. The red line shows the maximum intake recommended by the WHO.

**Table 1 foods-10-02452-t001:** TFA Country Score Card developed by the WHO [144].

Score	Countries	N of Countries
1. National policy commitment to eliminate TFA: National policies, strategies, or action plans that express a commitment to reduce iTFA in the food supply	Afghanistan, Albania, Algeria, Antigua and Barbuda, Bahamas, Bangladesh, Barbados, Belize, Benin, Bhutan, Bosnia and Herzegovina, Botswana, Cambodia, Chad, Costa Rica, Côte d’Ivoire, Djibouti, Dominican Republic, Egypt, Eswatini, French Polynesia, Ghana, Grenada, Guatemala, Guyana, Indonesia, Jamaica, Kenya, Lao People’s Democratic Republic, Lebanon, Maldives, Mauritania, Mauritius, Mongolia, Morocco, Myanmar, Namibia, Nauru, Nepal, Nigeria, North Macedonia, Papua New Guinea, Qatar, Saint Kitts and Nevis, Saint Lucia, Saint Vincent and the Grenadines, Samoa, Seychelles, Sri Lanka, Suriname, Timor-Leste, Trinidad and Tobago, Turkmenistan, Ukraine, United Arab Emirates, Vanuatu, Venezuela (Bolivarian Republic of), West Bank and Gaza Strip, Zambia	59
2. Other complementary measures: Legislative or other measures that encourage consumers to make healthier choices about iTFA or mandatory limits on iTFA in foods in specific settings	Bolivia (Plurinational State of), Brazil, Brunei Darussalam, Cape Verde, China, El Salvador, Ethiopia, Fiji, Israel, Jordan, Mexico, Oman, Pakistan, Paraguay, Philippines, Republic of Korea, Republic of Moldova, Tajikistan, Tunisia	20
3. Less restrictive TFA limits: Legislative or regulatory measures that limit iTFA in foods in all settings, but are less restrictive than the recommended approach	Argentina, Armenia, Bahrain, Belarus, Colombia, Ecuador, Georgia, India, Iran, Kazakhstan, Kuwait, Kyrgyzstan, Peru, Russian Federation, Singapore, Switzerland, Uruguay, Uzbekistan	18
4. Best-practice TFA policy: Legislative or regulatory measures that limit industrially produced TFA in foods in all settings, and are in line with the recommended approach	Austria, Belgium, Bulgaria, Canada, Chile, Croatia, Cyprus, Czechia, Denmark, Estonia, Finland, France, Germany, Greece, Guam, Hungary, Iceland, Ireland, Italy, Latvia, Liechtenstein, Lithuania, Luxembourg, Malta, Netherlands, Northern Mariana Islands, Norway, Poland, Portugal, Romania, Saudi Arabia, Slovakia, Slovenia, South Africa, Spain, Sweden, Thailand, United Kingdom of Great Britain and Northern Ireland, US	40
Best practice TFA policy passed but not yet in effect	Brazil, India, Paraguay, Peru, Singapore, Uruguay	6
Monitoring mechanism for mandatory TFA limits	Argentina, Armenia, Austria, Belarus, Canada, Chile, Colombia, Denmark, Ecuador, Georgia, Hungary, Iceland, India, Kazakhstan, Kyrgyzstan, Latvia, Lithuania, Norway, Peru, Russian Federation, Saudi Arabia, Singapore, South Africa, Switzerland, Thailand, US, Uruguay	27

**Table 2 foods-10-02452-t002:** Proportion of coronary heart disease death (%) due to TFA intake (>0.5% energy).

EU				EAEU	
Country	Proportion of CHD Death (%) Due to TFA Intake	Country	Proportion of CHD Death (%) Due to TFA Intake (>0.5% Energy)	Country	Proportion of CHD Death (%) Due to TFA Intake (>0.5% Energy)
Netherlands	14.4	Denmark	4.4	Belarus	5.8
Slovenia	7	Estonia	4.3	Kazakhstan	4.4
Austria	6.6	France	4.1	Russian Federation	4.3
Latvia	6.2	Croatia	4.1	Armenia	4.1
Czech Republic	5.9	Bulgaria	4	Kyrgyzstan	3.3
Germany	5.6	Belgium	4		
Poland	5.6	Ireland	3.9		
Lithuania	5.5	Romania	3.7		
Hungary	5.2	Malta	3.3		
Slovakia	5	Italy	3		
Luxembourg	4.8	Sweden	2.3		
Spain	4.7	Finland	2.7		
Portugal	4.6	Cyprus	1		
Greece	4.6				
Average				Average	
6.12				4.38	

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
