# Peer review of "The Effect of Trans Fatty Acids on Human Health: Regulation and Consumption Patterns"

_foods, 2021, doi:10.3390/foods10102452_

Round 1

Reviewer 1 Report

Minor point

L118-119: MFD is a syndrome in cows, not in humans. It is therefore unnecessary to mention this syndrome in the text.

Author Response

L118-119: MFD is a syndrome in cows, not in humans. It is therefore unnecessary to mention this syndrome in the text.

AUTHORS

We thank the reviewer for pointing out this mistake left over from previous versions in which this had been treated. Now we have removed it, thank you again.

Reviewer 2 Report

The paper has improved by the incorporation of more recent papers and by removing old ones.

I have detected that some sentences are not supported by the right references. For instance, reference 8 for sentence in line 943 is not correct. In that paper no discussion is made about the level of TFA in different foods. Some similar cases are the following:

- Ref 18 for line 952

- Ref 149 for line 948

- Ref 143, 154 for figure 3.

These cases, are at first sight, easily detected, but this makes me doubt about the rest. Authors should make a careful revision of the reference list and assure correct assignation.

There are a number of references that do not follow consecutive numbers. In line 732, reference 1123 is mentioned, when it should be 123. In line 747, ref 1125 is indicated. And it happens with references 1126, 1127, 1128, 1129, 1130, 1131…

Author Response

We thank the reviewer for the careful review, an incorrect numeration had been carried out and now corrected for all the references throughout the text.

This manuscript is a resubmission of an earlier submission. The following is a list of the peer review reports and author responses from that submission.

Round 1

Reviewer 1 Report

The subject of this review is interesting and concerns the health effects of trans fatty acids according to their industrial or natural origin. However, this review has many limitations.

Major points

This article is too long and not easy to read. It should be more synthesized, especially the introduction.

Method (section 2): The methodology is not adequate. In fact, many keywords are missing for the database search process, such as inflammation, insulin resistance, diabetes, oxidative stress, atherosclerosis. In addition, the words "trans-conjugated linoleic acid" are searched but not "conjugated linoleic acid", "conjugated linolenic acid" and "conjugated fatty acids". Thus this review is not exhaustive.

A PRISMA diagram must be made. Please indicate the number of articles obtained with these keywords and the articles obtained at the end. Was the quality of the articles also taken into account?

Discussion: The structure of the discussion is rather confusing. It would have been better to analyze for each biological parameter or pathology the beneficial and negative aspects. Indeed, the parameters presented are not the same in the Negative Health Effects and Beneficial Health Effects sections. In the Beneficial Health Effects section, only CLAs are discussed, which is worth noting.

Chapter 6: The first part of this chapter is not about the effects of TFAs on the gut microbiota but rather about the synthesis of CLAs by bacteria.

Figure 1: The contribution of different food groups to TFA intake is only for Anglo-Saxon countries. Why this choice? What about Spain, France or Italy? What about Eastern Europe? This is all the more unfortunate since the recommendations for Europe are discussed in Chapter 7. The reference for the American contribution is too old, the data should be updated.

Minor point

 It would be interesting to make a summary table of the amounts of trans fatty acids present in different foods.

The choice of references cited in the article is not always adequate. In particular, concerning the chemistry part, references 14 and 15 should be replaced by references concerning the structure and not the biological effect.

There are errors in Figure 2: O-H instead of OH; the cis configuration is not shown for linoleic acid. This should be corrected.

A85: please explain what milk fat depression is and give references for the whole sentence.

There is a paragraph 1.1 but where are paragraphs 1.2 and 2.1 and 2.2? This paragraph is too long and rather out of context.

A168, 169: references are missing.

Reviewer 2 Report

The review deals with a topic that has been extensively studied, as the authors state in their introduction. Many papers, some reviews included, have been published in the last 3 years about this topic, as you can see in the list I suggest below. These papers have not even mentioned in this manuscript. The originality of the review presented is lacking, and no novel contributions are seen.

See the list of relevant papers on this topic recently published:

Dietary Sources of Plasma trans Fatty Acids among Adults in the United States: NHANES 2009-2010. Li, Chaoyang; Richter, Patricia; Cobb, Laura K; Kuiper, Heather C; Seymour, Jennifer; Vesper, Hubert W. DOI:10.1093/cdn/nzab063

Ruminant and industrial trans-fatty acids consumption and cardiometabolic risk markers: A systematic review  Verneque, BJF ; Machado, AM ; Silva, LD; Lopes, ACS; Duarte, CK. CRITICAL REVIEWS IN FOOD SCIENCE AND NUTRITION DOI: 10.1080/10408398.2020.1836471

Industrially produced trans fat in popular foods in 15 countries of the former Soviet Union from 2015 to 2016: a market basket investigation. Stender, S. BMJ OPEN.  e023184 DOI: 10.1136/bmjopen-2018-023184

Developing a food composition table for estimating the intake of trans fatty acids. Miyazaki, S.; Matsumoto, Y.; Okada, C.; Kishida, T.; Nishioka, S.; Miyoshi, N.; Tomooka, K.; Tanigawa, T.; Saito, I.; Maruyama, K. Journal of Japanese Society of Nutrition and Food Science (Nippon Eiyo Shokuryo Gakkaishi). DOI:10.4327/jsnfs.74.93

Presence of trans fatty acids containing ingredients in pre-packaged foods in Australia in 2018. Huang, LP ; Federico, E; Jones, A; Wu, JHY.  AUSTRALIAN AND NEW ZEALAND JOURNAL OF PUBLIC HEALT.  DOI: 10.1111/1753-6405.13014

Mechanisms of Action of trans Fatty Acids. Oteng, AB ; Kersten, S. ADVANCES IN NUTRITION. DOI: 10.1093/advances/nmz125

Eighteen-carbon trans fatty acids and inflammation in the context of atherosclerosis. Valenzuela, CA; Baker, EJ; Miles, EA; Calder, PC. PROGRESS IN LIPID RESEARCH.  DOI: 10.1016/j.plipres.2019.101009

Global Surveillance of trans-Fatty Acids. Li, CY; Cobb, LK; Vesper, HW; Asma, PREVENTING CHRONIC DISEASE.  DOI: 10.5888/pcd16.190121

Impact of Danish ban of industrial produced trans fatty acids on serum cholesterol levels 1993-2006. Bjornsboe, K.; Jakobsen, M. U.; Bysted, A.; Fagt, S.; Christensen, T.; Joergensen, T. DOI:10.1093/eurpub/ckz185.387

Impact of Austria's 2009 trans fatty acids regulation on all-cause, cardiovascular and coronary heart disease mortality. Grabovac, I; Hochfellner, L; Rieger, M (; Jewell, J; Snell, A; Weber, A; Stuger, HP ; Schindler,; Mikkelsen,B. ] ; Dorner, TE. EUROPEAN JOURNAL OF PUBLIC HEALTH. DOI: 10.1093/eurpub/cky147

Progress towards elimination of trans-fatty acids in foods commonly consumed in four Latin American cities Monge-Rojas, R; Colon-Ramos, U; Jacoby, E; Alfaro, T; do Carmo, MDT  ; Villalpando, S] ; Bernal, C. PUBLIC HEALTH NUTRITION. DOI: 10.1017/S1368980017001227

Fatty acid profile of processed foods in Greece with focus on trans fatty acids Marakis, G; Fotakis, C; Tsigarida,; Mila, S ; Palilis,; Skoulika, S; Petropoulos, G; Papaioannou, A; Proestos, C. JOURNAL OF CONSUMER PROTECTION AND FOOD SAFETY. DOI: 10.1007/s00003-020-01290-1

Margarines and Fast-Food French Fries: Low Content of trans Fatty Acids.  Astiasaran, I; Abella, E; Gatta, G] ; Ansorena, D. NUTRIENTS. DOI: 10.3390/nu9070662

Fatty acid composition of sweet bakery goods and chocolate products and evaluation of overall nutritional quality in relation to the food label information. Omeroglu,; Ozdal, T.  JOURNAL OF FOOD COMPOSITION AND ANALYSIS. DOI: 10.1016/j.jfca.2020.103438

Some of these papers also demonstrate that the composition of foods has been changed during the last years, as a consequence of food policies aiming reducing trans fat from foods. Thus, old studies dealing with contributions to intakes do not reflect current situation in this topic. For instance, table 2, with data from 1996, or data from Mozzafarian et al., 2006 are not representative of current situation.  Bakery products, margarines, cookies have modified their composition… I would suggest authors to check for updated values and current literature that demonstrates the lack of these fatty acids in many reformulated products.

The discussion of the differences between industrial and ruminant fatty acids has also been widely debated, as some of the previous references point out, and this review does not contribute with new outcomes or conclusions.

Regarding rumen biohydrogenation, some of the references used are from 1997, 1977, 1975 and 1979. Some more recent papers such as DOI: 10.1099/mic.0.000811 and many others in this topic have been published.

Among the 148 references included in the paper, only one is from a paper published in 2021, only 4 from papers of 2020 and 4 from papers of 2019. This is again an indication of the poor literature search made.